# Scalable Rule-Based Representation Learning for Interpretable Classification

**Zhuo Wang**[1]**, Wei Zhang**[3]*****, Ning Liu**[4]**, Jianyong Wang**[1,2]*****

[1]Department of Computer Science and Technology, Tsinghua University
[2]Jiangsu Collaborative Innovation Center for Language Ability, Jiangsu Normal University
[3]School of Computer Science and Technology, East China Normal University
[4]School of Software, Shandong University
wang-z18@mails.tsinghua.edu.cn, {zhangwei.thu2011, victorliucs}@gmail.com,
jianyong@tsinghua.edu.cn

## Abstract

Rule-based models, e.g., decision trees, are widely used in scenarios demanding high model interpretability for their transparent inner structures and good model expressivity. However, rule-based models are hard to optimize, especially on large data sets, due to their discrete parameters and structures. Ensemble methods and fuzzy/soft rules are commonly used to improve performance, but they sacrifice the model interpretability. To obtain both good scalability and interpretability, we propose a new classifier, named Rule-based Representation Learner (RRL), that automatically learns interpretable non-fuzzy rules for data representation and classification. To train the non-differentiable RRL effectively, we project it to a continuous space and propose a novel training method, called Gradient Grafting, that can directly optimize the discrete model using gradient descent. An improved design of logical activation functions is also devised to increase the scalability of RRL and enable it to discretize the continuous features end-to-end. Exhaustive experiments on nine small and four large data sets show that RRL outperforms the competitive interpretable approaches and can be easily adjusted to obtain a trade-off between classification accuracy and model complexity for different scenarios. Our code is available at: `https://github.com/12wang3/rrl`.

## 1 Introduction

Although Deep Neural Networks (DNNs) have achieved impressive results in various machine learning tasks (Goodfellow et al., 2016), rule-based models, benefiting from their transparent inner structures and good model expressivity, still play an important role in domains demanding high model interpretability, such as medicine, finance, and politics (Doshi-Velez and Kim, 2017). In practice, rule-based models can easily provide explanations for users to earn their trust and help protect their rights (Molnar, 2019; Lipton, 2016). By analyzing the learned rules, practitioners can understand the decision mechanism of models and use their knowledge to improve or debug the models (Chu et al., 2018). Moreover, even if post-hoc methods can provide interpretations for DNNs, the interpretations from rule-based models are more faithful and specific (Murdoch et al., 2019). However, conventional rule-based models are hard to optimize, especially on large data sets, due to their discrete parameters and structures, which limit their application scope. To take advantage of rule-based models in more scenarios, we urgently need to answer such a question: *how to improve the scalability of rule-based models while keeping their interpretability?*

---

*****Corresponding authors.

35th Conference on Neural Information Processing Systems (NeurIPS 2021).

Studies in recent years provide some solutions to improve conventional rule-based models in different aspects. Ensemble methods and soft/fuzzy rules are proposed to improve the performance and scalability of rule-based models but at the cost of model interpretability (Ke et al., 2017; Breiman, 2001; Irsoy et al., 2012). Bayesian frameworks are also leveraged to more reasonably restrict and adjust the structures of rule-based models (Letham et al., 2015; Wang et al., 2017; Yang et al., 2017). However, due to the non-differentiable model structure, they have to use methods like MCMC or Simulated Annealing, which could be time-consuming for large models. Another way to improve rule-based models is to let a high-performance but complex model (e.g., DNN) teach a rule-based model (Frosst and Hinton, 2017; Ribeiro et al., 2016). However, learning from a complex model requires soft rules, or the fidelity of the student model is not guaranteed. The recent study Wang et al. (2020) tries to extract hierarchical rule sets from a tailored neural network. When the network is large, the extracted rules could behave quite differently from the neural network and become useless in most cases. Nevertheless, combined with binarized networks (Courbariaux et al., 2015), it inspires us that we can search for the discrete solutions of interpretable rule-based models in a continuous space leveraging effective optimization methods like gradient descent.

In this paper, we propose a novel rule-based model named **Rule-based Representation Learner (RRL)** (see Figure 1a). We summarize the key contributions as follows:

- To achieve good model **transparency and expressivity**, RRL is formulated as a hierarchical model, with layers supporting automatic feature discretization, rule-based representation learning in flexible conjunctive and disjunctive normal forms, and rule importance evaluation.
- To facilitate **training effectiveness**, RRL exploits a novel gradient-based discrete model training method, Gradient Grafting, that directly optimizes the discrete model and uses the gradient information at both continuous and discrete parametric points to accommodate more scenarios.
- To ensure **data scalability**, RRL utilizes improved logical activation functions to handle high-dimensional features. By further combining the improved logical activation functions with a tailored feature binarization layer, it realizes the continuous feature discretization end-to-end.
- We conduct experiments on nine small data sets and four large data sets to validate the advantages, i.e., good **accuracy and interpretability**, of our model over other representative classification models. The benefits of the model's key components are also verified by the experiments.

## 2   Related Work

**Rule-based Models**. Decision tree, rule list, and rule set are the widely used structures in rule-based models. For their discrete parameters and non-differentiable structures, we have to train them by employing various heuristic methods (Quinlan, 1993; Breiman, 2017; Cohen, 1995; Wei et al., 2019), which may not find the globally best solution or a solution with close performance. Alternatively, train them with search algorithms (Wang et al., 2017; Angelino et al., 2017; Lin et al., 2020), which could take too much time on large data sets. In recent studies, Bayesian frameworks are leveraged to restrict and adjust model structure more reasonably (Letham et al., 2015; Wang et al., 2017; Yang et al., 2017). Lakkaraju et al. (2016) learns independent if-then rules with smooth local search. However, except for heuristic methods, most existing rule-based models need frequent itemsets mining and/or long-time searching, which limits their applications. Moreover, it is hard for these rule-based models to get comparable performance with complex models like Random Forest.

Ensemble models like Random Forest (Breiman, 2001) and Gradient Boosted Decision Trees (Chen and Guestrin, 2016; Ke et al., 2017) have better performance than the single rule-based model. However, since the decision is made by hundreds of models, ensemble models are commonly not considered as interpretable models (Hara and Hayashi, 2016). Soft or fuzzy rules are also used to improve model performance (Irsoy et al., 2012; Ishibuchi and Yamamoto, 2005), but non-discrete rules are much harder to understand than discrete ones. Deep Neural Decision Tree (Yang et al., 2018) is a tree model realized by neural networks with the help of soft binning function and Kronecker product. However, due to the use of Kronecker product, it is not scalable with respect to the number of features. Other studies try to teach the rule-based model by a complex model, e.g., DNN, or extract rule-based models from complex models (Frosst and Hinton, 2017; Ribeiro et al., 2016; Wang et al., 2020). However, the fidelity of the student model or extracted model is not guaranteed.

**Gradient-based Discrete Model Training**. The gradient-based discrete model training methods are mainly proposed to train binary or quantized neural networks for network compression and

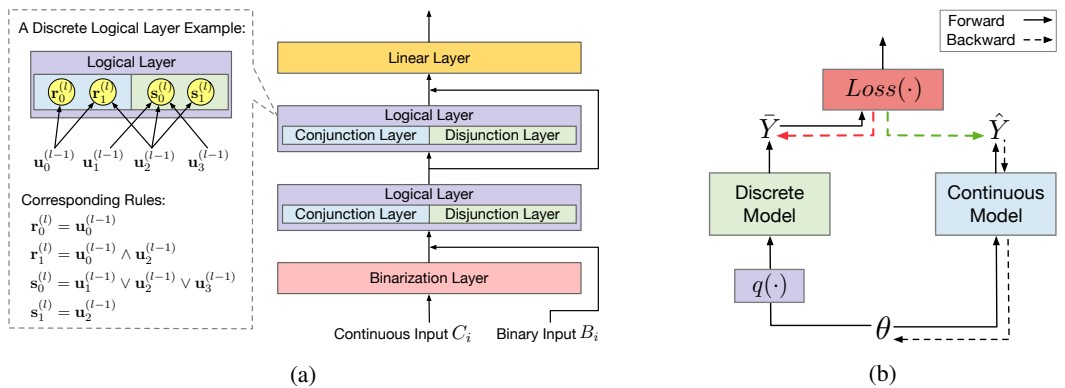

(a)                                                      (b)

Figure 1: (a) A Rule-based Representation Learner example. The dashed box shows an example of a discrete logical layer and its corresponding rules. (b) A simplified computation graph of Gradient Grafting. Arrows with solid lines represent forward pass while arrows with dashed lines represent backpropagation. The green arrow denotes the grafted gradient, a copy of the gradient represented by the red arrow. After grafting, there exists a backward path from the loss function to the parameter $\theta$.

acceleration. Courbariaux et al. (2015, 2016) propose to use the Straight-Through Estimator (STE) for binary neural network training. However, STE requires gradient information at discrete points, which limits its applications. ProxQuant (Bai et al., 2018) formulates quantized network training as a regularized learning problem and optimizes it via the prox-gradient method. ProxQuant can use gradients at non-discrete points but cannot directly optimize for the discrete model. The RB method (Wang et al., 2020) trains a neural network with random binarization for its weights to ensure the discrete and the continuous model behave similarly. However, when the model is large, the differences between the discrete and the continuous model are inevitable. Gumbel-Softmax estimator (Jang et al., 2016) generates a categorical distribution with a differentiable sample. However, it can hardly deal with a large number of variables, e.g., the weights of binary networks, for it is a biased estimator. Our method, i.e., Gradient Grafting, is different from all the aforementioned works in using gradient information of both discrete and continuous models in each backpropagation.

## 3 Rule-based Representation Learner

**Notation Description.** Let $\mathbb{D} = \{(X_1, Y_1), \ldots, (X_N, Y_N)\}$ denote the training data set with $N$ instances, where $X_i$ is the observed feature vector of the $i$-th instance with the $j$-th entry as $X_{i,j}$, and $Y_i$ is the corresponding one-hot class label vector, $i \in \{1, \ldots, N\}$. Each feature value can be either discrete or continuous. All the classes take discrete values, and the number of class labels is denoted by $M$. We use one-hot encoding to represent all discrete features as binary features. Let $C_i \in \mathbb{R}^m$ and $B_i \in \{0,1\}^b$ denote the continuous feature vector and the binary feature vector of the $i$-th instance respectively. Therefore, $X_i = C_i \oplus B_i$, where $\oplus$ represents the operator that concatenates two vectors. Throughout this paper, we use 1 (True) and 0 (False) to represent the two states of a Boolean variable. Thus each dimension of a binary feature vector corresponds to a Boolean variable.

**Overall Structure.** A Rule-based Representation Learner (RRL), denoted by $\mathcal{F}$, is a hierarchical model consisting of three different types of layers. Each layer in RRL not only contains a specific number of nodes but also has trainable edges connected with its previous layer. Let $\mathcal{U}^{(l)}$ denote the $l$-th layer of RRL, $\mathbf{u}_j^{(l)}$ indicate the $j$-th node in the layer, and $\boldsymbol{n}_l$ represent the corresponding number of nodes, $l \in \{0, \ldots, L\}$. The output of the $l$-th layer is a vector containing the values of all the nodes in the layer. For ease of expression, we denote this vector by $\mathbf{u}^{(l)}$. There are only one binarization layer, i.e., $\mathcal{U}^{(0)}$, and one linear layer, i.e., $\mathcal{U}^{(L)}$, in RRL, but the number of middle layers, i.e., logical layers, can be flexibly adjusted according to the specific situation. The logical layers mainly aim to learn the non-linear part of the data, while the linear layer aims to learn the linear part. One example of RRL is shown in Figure 1a.

When we input the $i$-th instance to RRL, the binarization layer will first binarize the continuous feature vector $C_i$ into a new binary vector $\bar{C}_i$. Then, $\bar{C}_i$ and $B_i$ are concatenated together as $\mathbf{u}^{(0)}$ and inputted to the first logical layer. The logical layers are designed to automatically learn data

representations using logical rules, and the stacked logical layers can learn rules in more complex forms. After going through all the logical layers, the output of the last logical layer can be considered as the new feature vector to represent the instance, wherein each feature corresponds to one rule formulated by the original features. As such, the whole RRL is composed of a feature learner and a linear classifier (linear layer). Moreover, the skip connections in RRL can skip unnecessary logical layers. In what follows, the details of these components will be elaborated.

### 3.1 Logical Layer

Considering the binarization layer needs the help of its following logical layer to binarize features in an end-to-end way, we introduce logical layers first. As mentioned above, logical layers can learn data representations using logical rules automatically. To achieve this, logical layers are designed to have a discrete version and a continuous version. The discrete version is used for training, testing and interpretation while the continuous version is only used for training. It is worth noting that the discrete RRL indicates the parameter weights of logical layers take discrete values (i.e., 0 or 1) while the parameter weights and biases of the linear layer still take continuous values.

**Discrete Version.** One logical layer consists of one conjunction layer and one disjunction layer. In discrete version, let $\mathcal{R}^{(l)}$ and $\mathcal{S}^{(l)}$ denote the conjunction and disjunction layer of $\mathcal{U}^{(l)}$ ($l \in \{1, 2, \ldots, L-1\}$) respectively. We denote the $i$-th node in $\mathcal{R}^{(l)}$ by $\mathbf{r}_i^{(l)}$, and the $i$-th node in $\mathcal{S}^{(l)}$ by $\mathbf{s}_i^{(l)}$. Specifically speaking, node $\mathbf{r}_i^{(l)}$ corresponds to the conjunction of nodes in the previous layer connected with $\mathbf{r}_i^{(l)}$, while node $\mathbf{s}_i^{(l)}$ corresponds to the disjunction of nodes in previous layer connected with $\mathbf{s}_i^{(l)}$. Formally, the two types of nodes are defined as follows:

$$\mathbf{r}_i^{(l)} = \bigwedge_{W_{i,j}^{(l,0)}=1} \mathbf{u}_j^{(l-1)}, \qquad \mathbf{s}_i^{(l)} = \bigvee_{W_{i,j}^{(l,1)}=1} \mathbf{u}_j^{(l-1)}, \tag{1}$$

where $W^{(l,0)}$ denote the adjacency matrix of the conjunction layer $\mathcal{R}^{(l)}$ and the previous layer $\mathcal{U}^{(l-1)}$, and $W_{i,j}^{(l,0)} \in \{0,1\}$. $W_{i,j}^{(l,0)} = 1$ indicates there exists an edge connecting $\mathbf{r}_i^{(l)}$ to $\mathbf{u}_j^{(l-1)}$, otherwise $W_{i,j}^{(l,0)} = 0$. Similarly, $W^{(l,1)}$ is the adjacency matrix of the disjunction layer $\mathcal{S}^{(l)}$ and $\mathcal{U}^{(l-1)}$. Similar to neural networks, we regard these adjacency matrices as the weight matrices of logical layers. $\mathbf{u}^{(l)} = \mathbf{r}^{(l)} \oplus \mathbf{s}^{(l)}$, where $\mathbf{r}^{(l)}$ and $\mathbf{s}^{(l)}$ are the outputs of $\mathcal{R}^{(l)}$ and $\mathcal{S}^{(l)}$ respectively.

The function of the logical layer is similar to the notion "level" in Wang et al. (2020). However, one level in that work, which actually consists of two layers, can only represent rules in Disjunctive Normal Form (DNF), while two logical layers in RRL can represent rules in DNF and Conjunctive Normal Form (CNF) at the same time. Connecting nodes in $\mathcal{R}^{(l)}$ with nodes in $\mathcal{S}^{(l-1)}$, we get rules in CNF, while connecting nodes in $\mathcal{S}^{(l)}$ with nodes in $\mathcal{R}^{(l-1)}$, we get rules in DNF. The flexibility of logical layer is quite important. For instance, the length of CNF rule $(\boldsymbol{a}_1 \vee \boldsymbol{a}_2) \wedge \cdots \wedge (\boldsymbol{a}_{2n-1} \vee \boldsymbol{a}_{2n})$ is $2n$, but the length of its corresponding DNF rule $(\boldsymbol{a}_1 \wedge \boldsymbol{a}_3 \cdots \wedge \boldsymbol{a}_{2n-1}) \vee \cdots \vee (\boldsymbol{a}_2 \wedge \boldsymbol{a}_4 \cdots \wedge \boldsymbol{a}_{2n})$ is $n \cdot 2^n$, which means layers that only represent DNF can hardly learn this CNF rule.

**Continuous Version.** Although the discrete logical layers have good interpretability, they are hard to train for their discrete parameters and non-differentiable structures. Inspired by the training process of binary neural networks that searches the discrete solution in a continuous space, we extend the discrete logical layer to a continuous version. The continuous version is differentiable, and when we discretize the parameters of a continuous logical layer, we can obtain its corresponding discrete logical layer. Therefore, the continuous logical layer and its corresponding discrete logical layer can also be considered as sharing the same parameters, but the discrete logical layer needs to discretize the parameters first.

Let $\hat{\mathcal{U}}^{(l)}$ denote the continuous logical layer, and $\hat{\mathcal{R}}^{(l)}$ and $\hat{\mathcal{S}}^{(l)}$ denote the continuous conjunction and disjunction layer respectively, $l \in \{1, 2, \ldots, L-1\}$. Let $\hat{W}^{(l,0)}$ and $\hat{W}^{(l,1)}$ denote the weight matrices of $\hat{\mathcal{R}}^{(l)}$ and $\hat{\mathcal{S}}^{(l)}$ respectively. $\hat{W}_{i,j}^{(l,0)}, \hat{W}_{i,j}^{(l,1)} \in [0,1]$. To make the whole Equation 1 differentiable, we leverage the logical activation functions proposed by Payani and Fekri (2019):

$$Conj(\mathbf{h}, W_i) = \prod_{j=1}^{n} F_c(\mathbf{h}_j, W_{i,j}), \qquad Disj(\mathbf{h}, W_i) = 1 - \prod_{j=1}^{n} (1 - F_d(\mathbf{h}_j, W_{i,j})), \tag{2}$$

where $F_c(h, w) = 1 - w(1 - h)$ and $F_d(h, w) = h \cdot w$. In Equation 2, if $\mathbf{h}$ and $W_i$ are both binary vectors, then $Conj(\mathbf{h}, W_i) = \bigwedge_{W_{i,j}=1} \mathbf{h}_j$ and $Disj(\mathbf{h}, W_i) = \bigvee_{W_{i,j}=1} \mathbf{h}_j$. $F_c(h, w)$ and $F_d(h, w)$ decide how much $\mathbf{h}_j$ would affect the operation according to $W_{i,j}$. If $W_{i,j} = 0$, $\mathbf{h}_j$ would have no effect on the operation. Equation 2 can be considered as extensions to t-norm fuzzy logic (Hájek, 2013). After using continuous weights and logical activation functions, the nodes in $\hat{\mathcal{R}}^{(l)}$ and $\hat{\mathcal{S}}^{(l)}$ are defined as follows:

$$\hat{\mathbf{r}}_i^{(l)} = Conj(\hat{\mathbf{u}}^{(l-1)}, \hat{W}_i^{(l,0)}), \qquad \hat{\mathbf{s}}_i^{(l)} = Disj(\hat{\mathbf{u}}^{(l-1)}, \hat{W}_i^{(l,1)}) \tag{3}$$

Now the whole logical layer is differentiable and can be trained by gradient descent. However, the above logical activation functions suffer from the serious vanishing gradient problem. The main reason can be found by analyzing the partial derivative of each node w.r.t. its directly connected weights and w.r.t. its directly connected nodes as follows:

$$\frac{\partial \hat{\mathbf{r}}_i^{(l)}}{\partial \hat{W}_{i,j}^{(l,0)}} = (\hat{\mathbf{u}}_j^{(l-1)} - 1) \cdot \prod_{k \neq j} F_c(\hat{\mathbf{u}}_k^{(l-1)}, \hat{W}_{i,k}^{(l,0)}), \qquad \frac{\partial \hat{\mathbf{r}}_i^{(l)}}{\partial \hat{\mathbf{u}}_j^{(l-1)}} = \hat{W}_{i,j}^{(l,0)} \cdot \prod_{k \neq j} F_c(\hat{\mathbf{u}}_k^{(l-1)}, \hat{W}_{i,k}^{(l,0)}) \tag{4}$$

Since $\hat{\mathbf{u}}_k^{(l-1)}$ and $\hat{W}_{i,k}^{(l,0)}$ are in the range $[0, 1]$, the values of $F_c(\cdot)$ in Equation 4 are in the range $[0, 1]$ as well. If $\mathbf{n}_{l-1}$ is large and most of the values of $F_c(\cdot)$ are not 1, then the derivative is close to 0 due to the multiplications (See Appendix D for the analysis of $\hat{\mathbf{s}}_i^{(l)}$). Wang et al. (2020) tries to use weight initialization to make $F_c(\cdot)$ close to 1 at the beginning. However, when dealing with hundreds of features, the vanishing gradient problem is still inevitable.

**Improved Logical Activation Functions.** We found that using the multiplications to simulate the logical operations in Equation 2 is the main reason for vanishing gradients and propose an improved design of logical activation functions. One straightforward idea is to convert multiplications into additions using logarithm, e.g., $\log(\prod_{j=1}^{n} F_c(\mathbf{h}_j, W_{i,j})) = \sum_{j=1}^{n} \log(F_c(\mathbf{h}_j, W_{i,j}))$. However, after taking the logarithm, the logical activation functions in Equation 2 cannot keep the characteristics of logical operations any more, and the ranges of $Conj(\cdot)$ and $Disj(\cdot)$ are not $[0, 1]$. To deal with this problem, we need a projection function to fix it. Apparently, the inverse function of $\log(x)$, i.e., $e^x$, is not suitable.

For the projection function $g$, three conditions must be satisfied: (i) $g(0) = e^0$. (ii) $\lim_{x \to -\infty} g(x) = \lim_{x \to -\infty} e^x = 0$. (iii) $\lim_{x \to -\infty} \frac{e^x}{g(x)} = 0$. Condition (i) and (ii) aim to keep the range and tendency of logical activation functions. Condition (iii) aims to lower the speed of approaching zero when $x \to -\infty$. In this work, we choose $g(x) = \frac{-1}{-1+x}$ as the projection function, and the improvement of logical activation functions can be summarized as the function $\mathbb{P}(v) = \frac{-1}{-1+\log(v)}$. The improved conjunction function $Conj_+$ and disjunction function $Disj_+$ are given by:

$$Conj_+(\mathbf{h}, W_i) = \mathbb{P}(\prod_{j=1}^{n}(F_c(\mathbf{h}_j, W_{i,j}) + \epsilon)), \quad Disj_+(\mathbf{h}, W_i) = 1 - \mathbb{P}(\prod_{j=1}^{n}(1 - F_d(\mathbf{h}_j, W_{i,j}) + \epsilon)) \tag{5}$$

where $\epsilon$ is a small constant, e.g., $10^{-10}$. The improved logical activation functions can avoid the vanishing gradient problem in most scenarios and are much more scalable than the originals. Moreover, considering that $\frac{d\mathbb{P}(v)}{dv} = \frac{\mathbb{P}^2(v)}{v}$, when $n$ in Equation 5 is extremely large, $\frac{d\mathbb{P}(v)}{dv}$ may be very close to 0 due to $\mathbb{P}^2(v)$. One trick to deal with it is replacing $\frac{\mathbb{P}^2(v)}{v}$ with $\frac{\mathbb{P}(\mathbb{P}^2(v))}{v}$ for $\mathbb{P}$ can lower the speed of approaching 0 while keeping the value range and tendency.

## 3.2 Binarization Layer

The binarization layer is mainly used to divide the continuous feature values into several bins. By combining one binarization layer and one logical layer, we can automatically choose the appropriate bins for feature discretization (binarization), i.e., binarizing features in an end-to-end way.

For the $j$-th continuous feature to be binarized, there are $k$ lower bounds ($\mathcal{T}_{j,1}, \ldots, \mathcal{T}_{j,k}$) and $k$ upper bounds ($\mathcal{H}_{j,1}, \ldots, \mathcal{H}_{j,k}$). All these bounds are randomly selected (e.g., from uniform distribution) in the value range of the $j$-th continuous feature, and these bounds are not trainable. When inputting one continuous feature vector $\mathbf{c}$, the binarization layer will check if $\mathbf{c}_j$ satisfies the bounds and get the following binary vector:

$$Q_j = [q(\mathbf{c}_j - \mathcal{T}_{j,1}), \ldots, q(\mathbf{c}_j - \mathcal{T}_{j,k}), q(\mathcal{H}_{j,1} - \mathbf{c}_j), \ldots, q(\mathcal{H}_{j,k} - \mathbf{c}_j)], \tag{6}$$

where $q(x) = \mathbf{1}_{x>0}$. If we input the $i$-th instance, i.e., $\boldsymbol{c} = C_i$, then $\bar{C}_i = Q_1 \oplus Q_2 \cdots \oplus Q_m$ and $\mathbf{u}^{(0)} = \bar{C}_i \oplus B_i$. After inputting $\mathbf{u}^{(0)}$ to the logical layer $\mathcal{U}^{(1)}$, the edge connections between $\mathcal{U}^{(1)}$ and $\mathcal{U}^{(0)}$ indicate the choice of bounds (bins). For example, if $\mathbf{r}_i^{(1)}$ is connected to the nodes corresponding to $q(\boldsymbol{c}_j - \mathcal{T}_{j,1})$ and $q(\mathcal{H}_{j,2} - \boldsymbol{c}_j)$, then $\mathbf{r}_i^{(1)}$ contains the bin $(\mathcal{T}_{j,1} < \boldsymbol{c}_j) \wedge (\boldsymbol{c}_j < \mathcal{H}_{j,2})$. If we replace $\mathbf{r}_i^{(1)}$ with $\mathbf{s}_i^{(1)}$ in the example, we can get $(\mathcal{T}_{j,1} < \boldsymbol{c}_j) \vee (\boldsymbol{c}_j < \mathcal{H}_{j,2})$. It should be noted that, in practice, if $\mathcal{T}_{j,1} \geq \mathcal{H}_{j,2}$, then $\mathbf{r}_i^{(1)} = 0$, and if $\mathcal{T}_{j,1} < \mathcal{H}_{j,2}$, then $\mathbf{s}_i^{(1)} = 1$. When using the continuous version, the weights of logical layers are trainable, which means we can choose bounds in an end-to-end way. For the number of bounds is $2k$ times of features, which could be large, only logical layers with improved logical activation functions are capable of choosing the bounds.

### 3.3 Gradient Grafting

Although RRL can be differentiable with the continuous logical layers, it is challenging to search for a discrete solution in a continuous space (Qin et al., 2020). One commonly used method to tackle this problem is the Straight-Through Estimator (STE) (Courbariaux et al., 2016). The STE method needs gradients at discrete points to update the parameters. However, the gradients of RRL at discrete points have no useful information in most cases (See Appendix E). Therefore STE is not suitable for RRL. Other methods like ProxQuant (Bai et al., 2018) and Random Binarization (Wang et al., 2020) cannot directly optimize for the discrete model and be scalable at the same time.

Inspired by plant grafting, we propose a new training method, called Gradient Grafting, that can effectively train RRL. In stem grafting, one plant is selected for its roots, i.e., rootstock, and the other plant is selected for its stems, i.e., scion. By grafting, we obtain a new plant with the advantages of both two plants. In Gradient Grafting, the gradient of the loss function w.r.t. the output of discrete model is the scion, and the gradient of the output of continuous model w.r.t. the parameters of continuous model is the rootstock. Specifically, let $\theta$ denote the parameter vector and $\theta^t$ denote the parameter vector at step $t$. $q(\boldsymbol{x}) = \mathbf{1}_{\boldsymbol{x}>0.5}$ is the binarization function that binarizes each dimension of $\boldsymbol{x}$ with 0.5 as the threshold. Let $\hat{Y}$ and $\bar{Y}$ denote the output of the continuous model $\hat{\mathcal{F}}$ and discrete model $\mathcal{F}$ respectively, then $\hat{Y} = \hat{\mathcal{F}}(\theta^t, X)$, $\bar{Y} = \mathcal{F}(q(\theta^t), X)$. The parameters update with Gradient Grafting is formulated by:

$$\theta^{t+1} = \theta^t - \eta \frac{\partial \mathcal{L}(\bar{Y})}{\partial \bar{Y}} \cdot \frac{\partial \hat{Y}}{\partial \theta^t}, \tag{7}$$

where $\eta$ is the learning rate and $\mathcal{L}(\cdot)$ is the loss function. One simplified computation graph of Gradient Grafting is shown in Figure 1b for intuitive understanding.

Gradient Grafting can directly optimize the loss of discrete models and use the gradient information at both continuous and discrete points, which overcomes the problems occurring in RRL training when using other gradient-based discrete model training methods. The convergence of Gradient Grafting is verified in the experiments (See Figure 3).

### 3.4 Model Interpretation

After training with Gradient Grafting, the discrete RRL can be used for testing and interpretation. RRL is easy to interpret, for we can simply consider it as a feature learner and a linear classifier. The binarization layer and logical layers are the feature learner, and they use logical rules to build and describe the new features. The linear classifier, i.e., the linear layer, makes decisions based on the new features. We can first find the important new features by the weights of the linear layer, then understand each new feature by analyzing its corresponding rule. One advantage of RRL is that it can be easily adjusted by the practitioners to obtain a trade-off between the classification accuracy and model complexity. Therefore, RRL can satisfy the requirements from different tasks and scenarios. There are several ways to limit the model complexity of RRL. First, we can reduce the number of logical layers in RRL, i.e., the depth of RRL, and the number of nodes in each logical layer, i.e., the width of RRL. Second, the L1/L2 regularization can be used during training to search for an RRL with shorter rules. The coefficient of the regularization term in the loss function can be considered as a hyperparameter to restrict the model complexity. After training, the dead nodes detection and redundant rules elimination proposed by Wang et al. (2020) can also be used for better interpretability.

Table 1: 5-fold cross validated F1 score (%) of comparing models on all 13 datasets. * represents that RRL significantly outperforms all the compared interpretable models (t-test with p < 0.01). The first seven models are interpretable models, while the last five are complex models.

| Dataset | RRL | C4.5 | CART | SBRL | CORELS | CRS | LR | SVM | PLNN(MLP) | RF | LightGBM | XGBoost |
|---|---|---|---|---|---|---|---|---|---|---|---|---|
| adult | 80.72 | 77.77 | 77.06 | 79.88 | 70.56 | **80.95** | 78.43 | 63.63 | 73.55 | 79.22 | 80.36 | 80.64 |
| bank-marketing | **76.32*** | 71.24 | 71.38 | 72.67 | 66.86 | 73.34 | 69.81 | 66.78 | 72.40 | 72.67 | 75.28 | 74.71 |
| banknote | **100.00*** | 98.45 | 97.85 | 94.44 | 98.49 | 94.93 | 98.82 | 100.00 | 100.00 | 99.40 | 99.48 | 99.55 |
| chess | 78.83 | 79.90 | 79.15 | 26.44 | 24.86 | 80.21 | 33.06 | 79.58 | 77.85 | 75.00 | 80.58 | **80.66** |
| connect-4 | **71.23*** | 61.66 | 61.24 | 48.54 | 51.72 | 65.88 | 49.87 | 69.85 | 64.55 | 62.72 | 70.53 | 70.65 |
| letRecog | 96.15* | 88.20 | 87.62 | 64.32 | 61.13 | 84.96 | 72.05 | 95.57 | 92.34 | **96.59** | 96.51 | 96.38 |
| magic04 | 86.33* | 82.44 | 81.20 | 82.52 | 77.37 | 80.87 | 75.72 | 79.43 | 83.07 | 86.48 | 86.67 | **86.69** |
| tic-tac-toe | **100.00** | 91.70 | 94.21 | 98.39 | 98.92 | 99.77 | 98.12 | 98.07 | 98.26 | 98.37 | 99.89 | 99.89 |
| wine | 98.23 | 95.48 | 94.39 | 95.84 | 97.43 | 97.78 | 95.16 | 96.05 | 76.07 | 98.31 | **98.44** | 97.78 |
| activity | 98.17 | 94.24 | 93.35 | 11.34 | 51.61 | 5.05 | 98.47 | 98.67 | 98.27 | 97.80 | **99.41** | 99.38 |
| dota2 | **60.12*** | 52.08 | 51.91 | 34.83 | 46.21 | 56.31 | 59.34 | 57.76 | 59.46 | 57.39 | 58.81 | 58.53 |
| facebook | **90.27*** | 80.76 | 81.50 | 31.16 | 34.93 | 11.38 | 88.62 | 87.20 | 89.43 | 87.49 | 85.87 | 88.90 |
| fashion | 89.01* | 80.49 | 79.61 | 47.38 | 38.06 | 66.92 | 84.53 | 84.46 | 89.36 | 88.35 | **89.91** | 89.82 |
| **AvgRank** | 2.77 | 8.23 | 8.92 | 9.31 | 9.92 | 7.08 | 7.92 | 6.77 | 5.77 | 5.38 | 2.77 | **2.69** |

# 4 Experiments

In this section, we conduct experiments to evaluate the proposed model and answer the following questions: (i) How is the classification performance of RRL? (ii) How is the model complexity of RRL? (iii) How is the convergence of Gradient Grafting compared to other gradient-based discrete model training methods? (iv) How is the scalability of the improved logical activation functions?

## 4.1 Dataset Description and Experimental Settings

**Dataset Description.** We took nine small and four large public datasets to conduct our experiments, all of which are often used to test classification performance and model interpretability (Dua and Graff, 2017; Xiao et al., 2017; Anguita et al., 2013; Rozemberczki et al., 2019). Appendix B summarizes the statistics of these 13 datasets. Together they show the data diversity, ranging from 178 to 102944 instances, from 2 to 26 classes, and from 4 to 4714 original features. See Appendix A for licenses.

**Performance Measurement.** We adopt the F1 score (Macro) as the classification performance metric since some of the data sets are imbalanced, i.e., the numbers of different classes are quite different. We adopt 5-fold cross-validation to evaluate the classification performance more fairly. The average rank of each model is also adopted for comparisons of classification performance over all the data sets (Demšar, 2006). Considering that reused structures exist in rule-based models, e.g., one branch in Decision Tree can correspond to several rules, we use the total number of edges instead of the total length of all rules as the metric of model complexity for rule-based models. See Appendix C for details about the **experiment environment** and **parameter settings** of all models.

## 4.2 Classification Performance

We compare the classification F1 score (Macro) of RRL with six interpretable models and five complex models. C4.5 (Quinlan, 1993), CART (Breiman, 2017), Scalable Bayesian Rule Lists (SBRL) (Yang et al., 2017), Certifiably Optimal Rule Lists (CORELS) (Angelino et al., 2017), and Concept Rule Sets (CRS) (Wang et al., 2020) are rule-based models. Logistic Regression (LR) (Kleinbaum et al., 2002) is a linear model. These six models are considered interpretable models. Piecewise Linear Neural Network (PLNN) (Chu et al., 2018), Support Vector Machines (SVM) (Scholkopf and Smola, 2001), Random Forest (Breiman, 2001), LightGBM (Ke et al., 2017), and XGBoost (Chen and Guestrin, 2016) are considered complex models. PLNN is a Multilayer Perceptron (MLP) that adopts piecewise linear activation functions, e.g., ReLU (Nair and Hinton, 2010). RF, LightGBM, and XGBoost are ensemble models. See Appendix C for the parameters tuning.

The results are shown in Table 1, and the first nine data sets are small data sets while the last four are large data sets. We can observe that RRL performs well on almost all the data sets and gets the best results on 6 data sets. The two-tailed Student's t-test (p<0.01) is used for significance testing, and we can observe that RRL significantly outperforms all the compared interpretable models on 8 out of 13 data sets. The average rank of RRL is also the top three among all the models. Only two complex models that use hundreds of estimators, i.e., XGBoost and LightGBM, have comparable results with

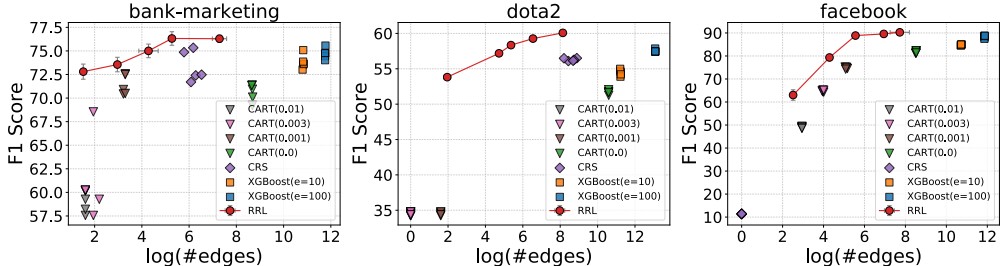

Figure 2: Scatter plot of F1 score against log(#edges) for RRL and baselines on three datasets (see Appendix K for other datasets).

RRL. For Table 1, the Friedman statistic is 73.84, corresponding to a p-value of $2.26 \times 10^{-11}$. Hence the null hypothesis of no significant differences is rejected. We then calculate pairwise comparisons using Conover post-hoc test, and RRL significantly outperforms all the compared interpretable models with p<0.01. Comparing RRL with LR and other rule-based models, we can see RRL can fit both linear and non-linear data well. CRS performs well on small data sets but fails on large datasets due to the limitation of its logical activation functions and training method. Good results on both small and large data sets verify RRL has good scalability. Moreover, SBRL and CRS do not perform well on continuous feature data sets like *letRecog* and *magic04* for they need preprocessing to discretize continuous features, which may bring bias to the data sets. On the contrary, RRL overcomes this problem by discretizing features end-to-end.

### 4.3 Model Complexity

Interpretable models seek to keep low model complexity while ensuring high accuracy. To show the relationships between accuracy and model complexity of RRL and baselines, we draw scatter plots of F1 score against log(#edges) for rule-based models in Figure 2 (see Appendix K for other data sets). The baselines are typical models in different categories of methods with good trade-offs between accuracy and model complexity. For RRL, the legend markers and error bars indicate means and standard deviations, respectively, of F1 score and log(#edges) across cross-validation folds. For baseline models, each point represents an evaluation of one model, on one fold, with one parameter setting. Therefore, in Figure 2, the closer its corresponding points are to the upper left corner, the better one model is. To obtain RRL with different model complexities, we tune the depth and width of RRL and the coefficient of L2 regularization term. The value in CART(·), e.g., CART(0.03), denotes the complexity parameter used for Minimal Cost-Complexity Pruning (Breiman, 2017), and a higher value corresponds to a simpler tree. We also show results of XGBoost with 10 and 100 estimators.

In Figure 2, on both small and large data sets, we can observe that if we connect the results of RRL, it will become a boundary that separating the upper left corner from other models. In other words, if RRL has a close model complexity with one baseline, then the F1 score of RRL will be higher. If the F1 score of RRL is close to one baseline, then the model complexity of RRL will be lower. It indicates that RRL can make better use of rules than rule-based models using heuristic and ensemble methods in most cases. The results in Figure 2 also verify that we can adjust the model complexity of RRL by setting the model structure and the coefficient of L2 regularization term. In this way, the practitioners are able to select an RRL with suitable classification performance and model complexity for different scenarios, which is crucial for practical applications of interpretable models.

### 4.4 Ablation Study

**Training Method for Discrete Model.** To show the effectiveness of Gradient Grafting for training RRL, we compare it with other representative gradient-based discrete model training methods, i.e., STE (Courbariaux et al., 2015, 2016), ProxQuant (Bai et al., 2018) and RB (Wang et al., 2020), by training RRL with the same structure. Hyperparameters are set to be the same for each method except exclusive hyperparameters, e.g., random binarization rate for RB, are fine-tuned. The training loss of the compared discrete model training methods and Gradient Grafting are shown in Figure 3, and we can see that the convergence of Gradient Grafting is faster and stabler than other methods on all data sets. As we mentioned in Section 3.3, RRL has little useful gradient information at discrete points,

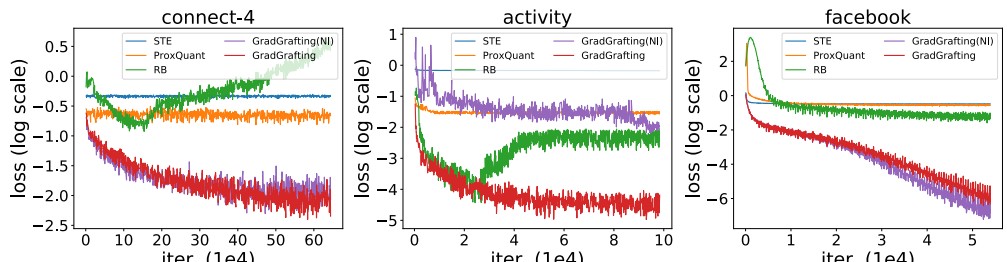

Figure 3: Training loss of three compared discrete model training methods and Gradient Grafting with or without improved logical activation functions on three data sets.

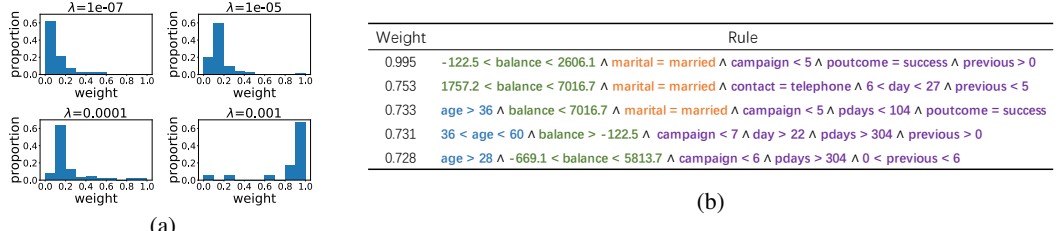

(a)                                                                                (b)

Figure 4: (a) The distribution of weights in the linear layer of RRLs trained on the *bank-marketing* data set with the same model structure but different $\lambda$, where $\lambda$ is the coefficient of the L2 regularization term. (b) Logical rules obtained from RRL trained on the *bank-marketing* data set.

thus RRL trained by STE cannot converge. Due to the difference between discrete and continuous RRL, RRL trained by ProxQuant and RB cannot converge well as well.

**Improved Logical Activation Functions.** We also compare RRL trained by Gradient Grafting with or without improved logical activation functions. The results are also shown in Figure 3, and GradGrafting(NI) represents RRL using original logical activation functions instead of improved logical activation functions. We can observe that the original activation functions work well on small data sets but fail on the large data set *activity* while the improved activation functions work well on all data sets, which means the improved logical activation functions make RRL more scalable. It should be noted that GradGrafting(NI) works well on the large data set *facebook*. The reason is *facebook* is a very sparse data set, and the number of 1 in each binary feature vector is less than 30 (See Appendix D for detailed analyses).

## 4.5 Case Study

We show how the learned RRL looks like by case studies. Take the *bank-marketing* data set as an example. We first show the distribution of weights in the linear layer of the trained RRLs in Figure 4a. Each weight in the linear layer corresponds to one rule. For a better viewing experience, we show the normalized absolute values of weights. The model structures of these RRLs are the same, but different coefficients of the L2 regularization term, denoted by $\lambda$, lead to different model complexities. We can observe that, when $\lambda$ is small, which means the RRL is more complex, there are many rules with small weights. These small weight rules are mainly used to slightly adjust the outputs. Hence, they make the RRL more accurate but less interpretable. However, in practice, we can ignore these small weight rules and only focus on rules with large weights first. After analyzing rules with large weights and having a better understanding of the learned RRL and the data set, we can then understand those less important rules gradually. When $\lambda$ is large, the number of rules is small, and we can directly understand the whole RRL rather than understanding RRL step by step.

In Figure 4b, we show the learned rules, with high weights, of one RRL trained on the *bank-marketing* data set (see Appendix J for the *fashion* data set). These rules are used to predict if the client will subscribe a term deposit by telesales. Different types of features are marked in different colors, e.g., purple for previous behaviours of the bank. We can clearly see that middle-aged married persons with low balance are more likely to subscribe a deposit, and the previous behaviour of the bank would also affect the client. Then the bank can change its strategies according to these rules.

# 5 Conclusion and Future Work

We propose a new scalable classifier, named Rule-based Representation Learner (RRL), that can automatically learn interpretable rules for data representation and classification. For the particularity of RRL, we propose a new gradient-based discrete model training method, i.e., Gradient Grafting, that directly optimizes the discrete model. We also propose an improved design of logical activation functions to increase the scalability of RRL and make RRL capable of discretizing the continuous features end-to-end. Our experimental results show that RRL enjoys both high classification performance and low model complexity on data sets with different scales. For the current design of RRL is limited to structured data, when dealing with unstructured data using RRL, we need to convert them into structured data first. For example, RRL needs to convert a 2D image into a 1D vector, which could lose spatial information. We will extend RRL to suit more unstructured data as future work.

# 6 Acknowledgments

This work was supported in part by National Key Research and Development Program of China under Grant No. 2020YFA0804503, National Natural Science Foundation of China under Grant No. 61521002 and 62072182, and Beijing Academy of Artificial Intelligence (BAAI).

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
