# A  Code Release and Data Source

Our code is publicly available at a GitHub repository: `https://github.com/12wang3/rrl`.

The datasets used in this paper come from the UCI machine learning repository and GitHub. The links to the datasets are: *adult*[2], *bank-marketing*[3], *banknote*[4], *chess*[5], *connect-4*[6], *letRecog*[7], *magic04*[8], *tic-tac-toe*[9], *wine*[10], *activity*[11], *dota2*[12], *facebook*[13], *fashion*[14]. The *fashion* dataset uses the MIT license, and other datasets have no license mentioned. The citation requests of these datasets are all satisfied in our paper.

# B  Data Sets Properties

In Table 2, the first nine data sets are small data sets while the last four are large data sets. Discrete or continuous feature type indicates features in that data set are all discrete or all continuous. The mixed feature type indicates the corresponding data set has both discrete and continuous features. The density is the averaged ratio of the number of 1 in each binary feature vector after one-hot encoding.

Table 2: Data sets properties.

| Dataset | #instances | #classes | #features | feature type | density |
|---------|-----------|----------|-----------|--------------|---------|
| adult | 32561 | 2 | 14 | mixed | - |
| bank-marketing | 45211 | 2 | 16 | mixed | - |
| banknote | 1372 | 2 | 4 | continuous | - |
| chess | 28056 | 18 | 6 | discrete | 0.150 |
| connect-4 | 67557 | 3 | 42 | discrete | 0.333 |
| letRecog | 20000 | 26 | 16 | continuous | - |
| magic04 | 19020 | 2 | 10 | continuous | - |
| tic-tac-toe | 958 | 2 | 9 | discrete | 0.333 |
| wine | 178 | 3 | 13 | continuous | - |
| activity | 10299 | 6 | 561 | continuous | - |
| dota2 | 102944 | 2 | 116 | discrete | 0.087 |
| facebook | 22470 | 4 | 4714 | discrete | 0.003 |
| fashion | 70000 | 4 | 784 | continuous | - |

# C  Experimental Setting

**Experiment Environment.** We implement our model with PyTorch (Paszke et al., 2019), an open-source machine learning framework. Experiments are conducted on a Linux server with an Intel Xeon E5 v4 CPU at 2.10GHz and one GeForce RTX 2080 Ti GPU.

**Parameter Settings.** The number of logical layers in RRL ranges from 1 to 3. The number of nodes in logical layers ranges from 16 to 4096 depending on the number of binary features of the data set and the model complexity we need. We use the cross-entropy loss during the training. The L2 regularization is adopted to restrict the model complexity, and the coefficient of the regularization

---

[2]`https://archive.ics.uci.edu/ml/datasets/Adult`

[3]`http://archive.ics.uci.edu/ml/datasets/Bank+Marketing`

[4]`https://archive.ics.uci.edu/ml/datasets/banknote+authentication`

[5]`https://archive.ics.uci.edu/ml/datasets/Chess+%28King-Rook+vs.+King%29`

[6]`http://archive.ics.uci.edu/ml/datasets/connect-4`

[7]`https://archive.ics.uci.edu/ml/datasets/Letter+Recognition`

[8]`https://archive.ics.uci.edu/ml/datasets/magic+gamma+telescope`

[9]`https://archive.ics.uci.edu/ml/datasets/Tic-Tac-Toe+Endgame`

[10]`https://archive.ics.uci.edu/ml/datasets/wine`

[11]`https://archive.ics.uci.edu/ml/datasets/human+activity+recognition+using+smartphones`

[12]`https://archive.ics.uci.edu/ml/datasets/Dota2+Games+Results`

[13]`https://archive.ics.uci.edu/ml/datasets/Facebook+Large+Page-Page+Network`

[14]`https://github.com/zalandoresearch/fashion-mnist`

term in the loss function is in $\{10^{-2}, 10^{-3}, \ldots, 10^{-8}, 0\}$. The numbers of the lower and upper bounds in the binarization layer are both in $\{5, 10, 50\}$. We utilize the Adam (Kingma and Ba, 2014) method for the training process with a mini-batch size of 32. The initial learning rate is in $\{5 \times 10^{-3}, 2 \times 10^{-3}, 1 \times 10^{-3}, 5 \times 10^{-4}, 2 \times 10^{-4}, 1 \times 10^{-4}\}$. On small data sets, RRL is trained for 400 epochs, and we decay the learning rate by a factor of 0.75 every 100 epochs. On large data sets, RRL is trained for 100 epochs, and we decay the learning rate by a factor of 0.75 every 20 epochs. We use the derivative estimation trick mentioned in Section 3.1 for RRL trained on large data sets. When parameter tuning is required, 95% of the training set is used for training and 5% for validation.

We use validation sets to tune hyperparameters of all the baselines mentioned in Section 4.2. We use sklearn to implement LR, and use the L1 or L2 norm in the penalization. The liblinear is used as the solver. The tolerance for stopping criteria is in $\{10^{-3}, 10^{-4}, 10^{-5}\}$. The inverse of regularization strength is in $\{1, 4, 16, 32\}$. For decision tree, its max depth is in {None, 5, 10, 20}. The min number of samples required to split an internal node is in {2, 8, 16}, and the min number of samples required to be at a leaf node is in {1, 8, 16}. For SBRL, its $\lambda$ is set to 5 initially, and the min and max rule sizes are set at 1 and 3, respectively. $\eta$ is set to 1, and the numbers of iterations and chains are set to 5000 and 20, respectively. The minsupport_pos and minsupport_neg are set to keep the total number of rules close to 300. For CORELS, the regularization parameter is in $\{0, 10^{-2}, 10^{-3}, 10^{-4}, 10^{-5}\}$, and the maximum number of rule lists to search is in $\{10^4, 10^5, 10^6, 10^7\}$. The maximum cardinality allowed is set to 2 or 3, and the min support rate is in {0.005, 0.01, 0.025, 0.05}. For CRS and PLNN, the candidate sets of learning rate, learning rate decay rate, batch size, model structure (depth and width), and coefficient of the regularization term in the loss function are the same as RRL's. The random binarization rate of CRS is in {0, 0.7, 0.75, 0.8, 0.85, 0.9, 0.95}. For SVM, the linear, RBF and Ploy kernels are used. The tolerance for stopping criteria is in $\{10^{-3}, 10^{-4}, 10^{-5}\}$. The inverse of regularization strength is in {1, 4, 16, 32}. The kernel coefficient is set to the reciprocal of the number of features. For Random Forest, its min number of samples required to split an internal node and the min number of samples required to be at a leaf node are the same as the decision tree's. We use LightGBM and XGBoost to implement Gradient Boosted Decision Tree (GBDT). The learning rate of GBDT is in {0.1, 0.01, 0.001}, and the max depth of one tree is in {None, 5, 10, 20}. The number of estimators in ensemble models is in {10, 100, 500}. For baselines that can not directly deal with the continuous value inputs, we use the recursive minimal entropy partitioning algorithm or the KBinsDiscretizer implemented by sklearn to discretize the inputs first. Grid search is also used for the parameter tuning.

## D   Vanishing Gradient Problem

The partial derivative of each node in $\hat{\mathcal{S}}^{(l)}$ w.r.t. its directly connected weights and w.r.t. its directly connected nodes are given by:

$$\frac{\partial \hat{\mathbf{s}}_i^{(l)}}{\partial \hat{W}_{i,j}^{(l,1)}} = \hat{\mathbf{u}}_j^{(l-1)} \cdot \prod_{k \neq j} (1 - F_d(\hat{\mathbf{u}}_k^{(l-1)}, \hat{W}_{i,k}^{(l,1)})) \tag{8}$$

$$\frac{\partial \hat{\mathbf{s}}_i^{(l)}}{\partial \hat{\mathbf{u}}_j^{(l-1)}} = \hat{W}_{i,j}^{(l,1)} \cdot \prod_{k \neq j} (1 - F_d(\hat{\mathbf{u}}_k^{(l-1)}, \hat{W}_{i,k}^{(l,1)})) \tag{9}$$

Similar to the analysis of Equation 4, due to $\hat{\mathbf{u}}_k^{(l-1)}$ and $\hat{W}_{i,k}^{(l,1)}$ are in the range $[0, 1]$, the values of $(1 - F_d(\cdot))$ in Equation 8 and 9 are in the range $[0, 1]$ as well. If $\mathbf{n}_{l-1}$ is large and most of the values of $(1 - F_d(\cdot))$ are not 1, then the derivative is close to 0 because of the multiplications. Therefore, both the conjunction function and the disjunction function suffer from vanishing gradient problem.

If the large data set is very sparse and the number of 1 in each binary feature vector (for RRL the binary feature vector is $\mathbf{u}^{(0)}$) is less than about one hundred, there will be no vanishing gradient problem for nodes in $\hat{\mathcal{S}}^{(1)}$. The reason is when the number of 1 in each feature vector is less than about one hundred, in Equation 8 and 9, most of the values of $(1 - F_d(\cdot))$ are 1, and only less than one hundred values of $(1 - F_d(\cdot))$ are not 1, then the result of the multiplication is not very close to 0. The *facebook* data set is an example of this case. However, if the number of 1 in each binary feature vector is more than about one hundred, the vanishing gradient problem comes again.

# E  Gradients at Discrete Points

The gradients of RRL with original logical activation functions at discrete points can be obtained by Equation 4, 8 and 9. Take Equation 8 as an example, discrete points mean all the weights of logical layers are 0 or 1, which also means the values of all the nodes in $\hat{\mathcal{U}}^{(l)}$ are 0 or 1, $l \in \{0, 1, \ldots, L-1\}$. Hence, in Equation 8, $\hat{\mathbf{u}}_j^{(l-1)}, (1 - F_d(\cdot)) \in \{0, 1\}$, and the whole equation is actually multiplications of several 0 and several 1. Only when $\hat{\mathbf{u}}_j^{(l-1)}$ and $(1 - F_d(\cdot))$ are all 1, the derivative in Equation 8 is 1, otherwise, the derivative is 0. Therefore, the gradients at discrete points have no useful information in most cases. The analyses of Equation 4 and 9 are similar.

To analyze the gradients of RRL with improved logical activation functions at discrete points, we first calculate the partial derivative of each node w.r.t. its directly connected weights and w.r.t. its directly connected nodes:

$$\frac{\partial \hat{\mathbf{r}}_i^{(l)}}{\partial \hat{W}_{i,j}^{(l,0)}} = \frac{(\hat{\mathbf{r}}_i^{(l)})^2}{F_c(\hat{\mathbf{u}}_j^{(l-1)}, \hat{W}_{i,j}^{(l,0)}) + \epsilon} \cdot (\hat{\mathbf{u}}_j^{(l-1)} - 1) \tag{10}$$

$$\frac{\partial \hat{\mathbf{r}}_i^{(l)}}{\partial \hat{\mathbf{u}}_j^{(l-1)}} = \frac{(\hat{\mathbf{r}}_i^{(l)})^2}{F_c(\hat{\mathbf{u}}_j^{(l-1)}, \hat{W}_{i,j}^{(l,0)}) + \epsilon} \cdot \hat{W}_{i,j}^{(l,0)} \tag{11}$$

$$\frac{\partial \hat{\mathbf{s}}_i^{(l)}}{\partial \hat{W}_{i,j}^{(l,1)}} = \frac{(1 - \hat{\mathbf{s}}_i^{(l)})^2}{1 - F_d(\hat{\mathbf{u}}_j^{(l-1)}, \hat{W}_{i,j}^{(l,1)}) + \epsilon} \cdot \hat{\mathbf{u}}_j^{(l-1)} \tag{12}$$

$$\frac{\partial \hat{\mathbf{s}}_i^{(l)}}{\partial \hat{\mathbf{u}}_j^{(l-1)}} = \frac{(1 - \hat{\mathbf{s}}_i^{(l)})^2}{1 - F_d(\hat{\mathbf{u}}_j^{(l-1)}, \hat{W}_{i,j}^{(l,1)}) + \epsilon} \cdot \hat{W}_{i,j}^{(l,1)} \tag{13}$$

Take Equation 10 for example, when all the weights of logical layers are 0 or 1, the $\hat{\mathbf{r}}_i^{(l)}$, $F_c(\cdot) + \epsilon$ and $(\hat{\mathbf{u}}_j^{(l-1)} - 1)$ are all very close to 0 or 1 as well. For the initialized weights are randomly selected, $\hat{\mathbf{r}}_i^{(l)}$ is close to 0 in most cases. Hence, the derivative in Equation 10 is close to 0 in most cases, and the analyses of Equation 11, 12 and 13 are similar. Therefore, the gradients at discrete points have little useful information.

# F  Computation Time

The computation time of RRL is similar to neural networks like Multilayer Perceptrons (MLP) for their computations are quite similar. The training time of RRL (in Table 1) on all the datasets (400 epochs on the small datasets and 100 epochs on the large datasets) with one GeForce RTX 2080 Ti is shown in Table 3. We can see that the training time of RRL is acceptable on all the datasets, which also verifies the good scalability of RRL.

Table 3: Training time of RRL on nine small and four large datasets.

| Dataset | adult | bank-marketing | banknote | chess | connect-4 | letRecog |
|---------|-------|----------------|----------|-------|-----------|----------|
| Time | 1h22m55s | 1h3m41s | 7m45s | 29m40s | 2h20m41s | 2h16m24s |
| Dataset | magic04 | tic-tac-toe | wine | activity | dota | facebook | fashion |
| Time | 3h22m37s | 1m12s | 16s | 1h2m24s | 1h58m42s | 2h27m23s | 7h32m52s |

# G  Rule Length

The average length of rules in RRL (in Table 1) trained on different datasets is shown in Table 4. We can observe that, except for the *facebook* and *fashion* datasets, the average length of rules is less than 13 (most are less than 7), which means understanding one rule is easy and understanding the rules one by one in the order of weights is feasible. The average length of rules of RRL trained on the

*facebook* and *fashion* datasets is large for *facebook* and *fashion* are actually two unstructured datasets, e.g., the *fashion* dataset is an image classification dataset.

Table 4: Average rule length of RRL trained on nine small and four large datasets.

| Dataset | adult | bank-marketing | banknote | chess | connect-4 | letRecog |
|---|---|---|---|---|---|---|
| AvgLength | 5.71 | 3.78 | 3.56 | 7.03 | 12.44 | 5.91 |

| Dataset | magic04 | tic-tac-toe | wine | activity | dota | facebook | fashion |
|---|---|---|---|---|---|---|---|
| AvgLength | 5.05 | 2.88 | 2.11 | 6.67 | 4.37 | 38.28 | 34.53 |

The minimum and maximum lengths of rules in RRL (in Table 1) trained on different datasets are shown in Table 5.

Table 5: Minimum and maximum rule lengths of RRL trained on nine small and four large datasets.

| Dataset | adult | bank-marketing | banknote | chess | connect-4 | letRecog |
|---|---|---|---|---|---|---|
| [Min, Max] | [1, 12] | [1, 13] | [2, 6] | [1, 16] | [1, 20] | [1, 12] |

| Dataset | magic04 | tic-tac-toe | wine | activity | dota | facebook | fashion |
|---|---|---|---|---|---|---|---|
| [Min, Max] | [1, 16] | [2, 4] | [1, 5] | [1, 17] | [1, 12] | [11, 119] | [1, 101] |

# H  Classification Performance

We also use accuracy to evaluate the classification performance, and other settings are the same as Section 4.2. The results are shown in Table 6.

Table 6: 5-fold cross validated accuracy (%) of comparing models on all 13 datasets. $*$ represents that RRL significantly outperforms all the compared interpretable models (t-test with $p < 0.01$). The first seven models are interpretable models, while the last five are complex models.

| Dataset | RRL | C4.5 | CART | SBRL | CORELS | CRS | LR | SVM | PLNN(MLP) | RF | LightGBM | XGBoost |
|---|---|---|---|---|---|---|---|---|---|---|---|---|
| adult | 85.73 | 85.28 | 85.19 | 84.83 | 79.98 | 85.71 | 85.25 | 79.78 | 85.71 | 85.57 | 85.96 | **86.13** |
| bank-marketing | 90.63 | 89.93 | 89.83 | 90.38 | 88.30 | 90.11 | 90.13 | 88.31 | 90.54 | 90.68 | 90.76 | **90.91** |
| banknote | **100.00**$^*$ | 98.61 | 98.03 | 93.95 | 97.08 | 93.46 | 98.83 | 100.00 | 100.00 | 99.56 | 99.64 | 99.56 |
| chess | 82.02 | 79.20 | 78.67 | 29.37 | 32.39 | 82.38 | 36.07 | 82.17 | 77.05 | 75.14 | 84.93 | **85.40** |
| connect-4 | **86.96**$^*$ | 76.32 | 76.38 | 62.80 | 64.93 | 79.05 | 75.74 | 83.65 | 85.39 | 83.10 | 85.53 | 86.47 |
| letRecog | 95.59$^*$ | 88.18 | 87.51 | 62.88 | 58.48 | 83.13 | 72.31 | 94.92 | 92.48 | 96.54 | **96.86** | 96.45 |
| magic04 | 87.52$^*$ | 84.77 | 83.72 | 83.58 | 79.09 | 83.81 | 79.02 | 82.68 | 86.49 | 88.20 | 88.46 | **88.62** |
| tic-tac-toe | 100.00 | 94.26 | 93.32 | 98.85 | 98.75 | 99.06 | 98.33 | 98.33 | 98.12 | 98.75 | 99.90 | 99.27 |
| wine | 97.76 | 93.83 | 90.46 | 92.70 | 94.52 | 97.22 | 95.54 | 91.03 | 90.52 | **97.78** | 96.08 | 96.06 |
| activity | 97.96 | 94.08 | 93.31 | 18.25 | 61.04 | 17.33 | 98.33 | 98.29 | 98.28 | 97.92 | **99.37** | 99.26 |
| dota2 | **59.87** | 52.92 | 52.73 | 51.91 | 53.02 | 55.95 | 59.46 | 59.57 | 55.27 | 58.20 | 58.80 | 59.35 |
| facebook | **90.53**$^*$ | 82.59 | 82.90 | 49.72 | 41.27 | 28.91 | 89.44 | 88.46 | 90.00 | 88.32 | 86.85 | 88.00 |
| fashion | 89.22$^*$ | 80.68 | 79.43 | 54.57 | 49.14 | 67.71 | 84.78 | 89.34 | 89.29 | 88.53 | **91.35** | 90.71 |
| **AvgRank** | 2.92 | 8.15 | 9.23 | 9.92 | 10.23 | 7.62 | 7.23 | 6.15 | 5.62 | 5.08 | **2.77** | **2.77** |

# I  Ablation Study

**Training Method for Discrete Model.** To show the effectiveness of Gradient Grafting for training RRL, we compare it with other representative gradient-based discrete model training methods, i.e., STE (Courbariaux et al., 2015, 2016), ProxQuant (Bai et al., 2018) and RB (Wang et al., 2020), by training RRL with the same structure. Hyperparameters are set to be the same for each method except exclusive hyperparameters, e.g., random binarization rate for RB, are fine-tuned. The training loss of the compared discrete model training methods and Gradient Grafting are shown in Figure 5, and we can see that the convergence of Gradient Grafting is faster and stabler than other methods on all data sets.

**Improved Logical Activation Functions.** We also compare RRL trained by Gradient Grafting with or without improved logical activation functions. The results are also shown in Figure 5, and

GradGrafting(NI) represents RRL using original logical activation functions instead of improved logical activation functions.

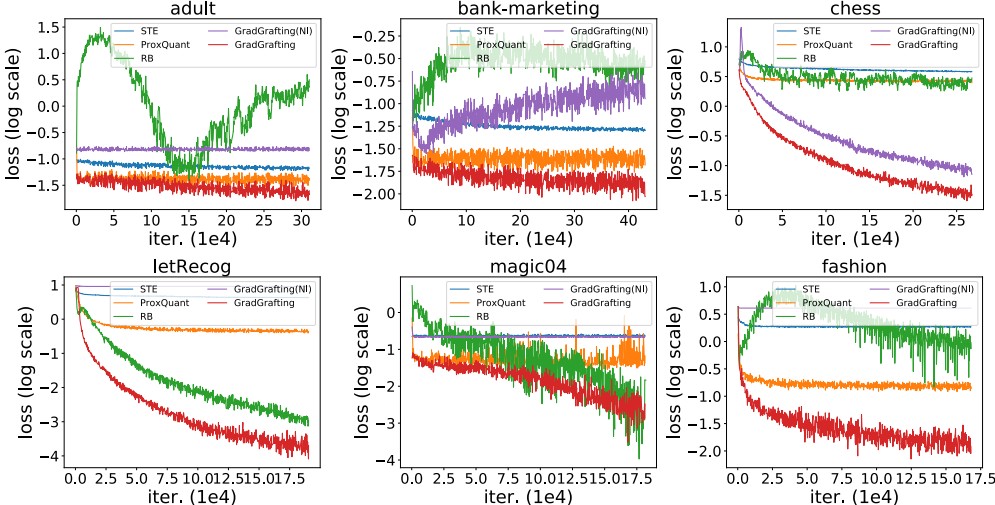

Figure 5: Training loss of three compared discrete model training methods and Gradient Grafting with or without improved logical activation functions on six data sets.

## J  Case Study

Although RRL is not designed for image classification tasks, due to its high scalability, it can still provide intuition by visualizations. Take the *fashion* dataset for example, for each class, we combine the first ten rules, ordered by linear layer weights, for feature (pixel) visualization. In Figure 6, a black/white pixel indicates the combined rule asks for a color close to black/white here in the original input image, and the grey pixel means no requirement in the rule. According to these figures, we can see how RRL classifies the images, e.g., distinguishing T-shirt from Pullover by sleeves.

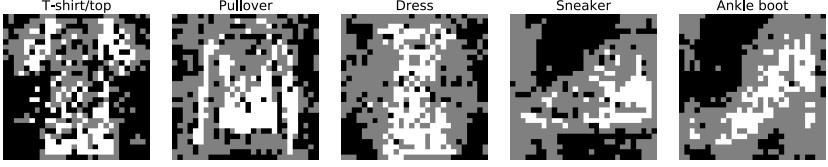

Figure 6: Decision mode for the *fashion* data set summarized from rules of RRL.

## K  Model Complexity

Figure 7 shows the scatter plots of F1 score against log(#edges) for rule-based models trained on the other ten data sets. For RRL, the legend markers and error bars indicate means and standard deviations, respectively, of F1 score and log(#edges) across cross-validation folds. For baseline models, each point represents an evaluation of one model, on one fold, with one parameter setting. The value in CART(·), e.g., CART(0.03), denotes the complexity parameter used for Minimal Cost-Complexity Pruning (Breiman, 2017), and a higher value corresponds to a simpler tree. We also show the results of XGBoost with 10 and 100 estimators. On these ten data sets, we can still observe that if we connect the results of RRL, it will become a boundary that separating the upper left corner from other models. In other words, if RRL has a close model complexity with one baseline, then the F1 score of RRL will be higher, or if the F1 score of RRL is close to one baseline, then its model complexity will be lower. It indicates that RRL can make better use of rules than rule-based models using heuristic and ensemble methods in most cases.

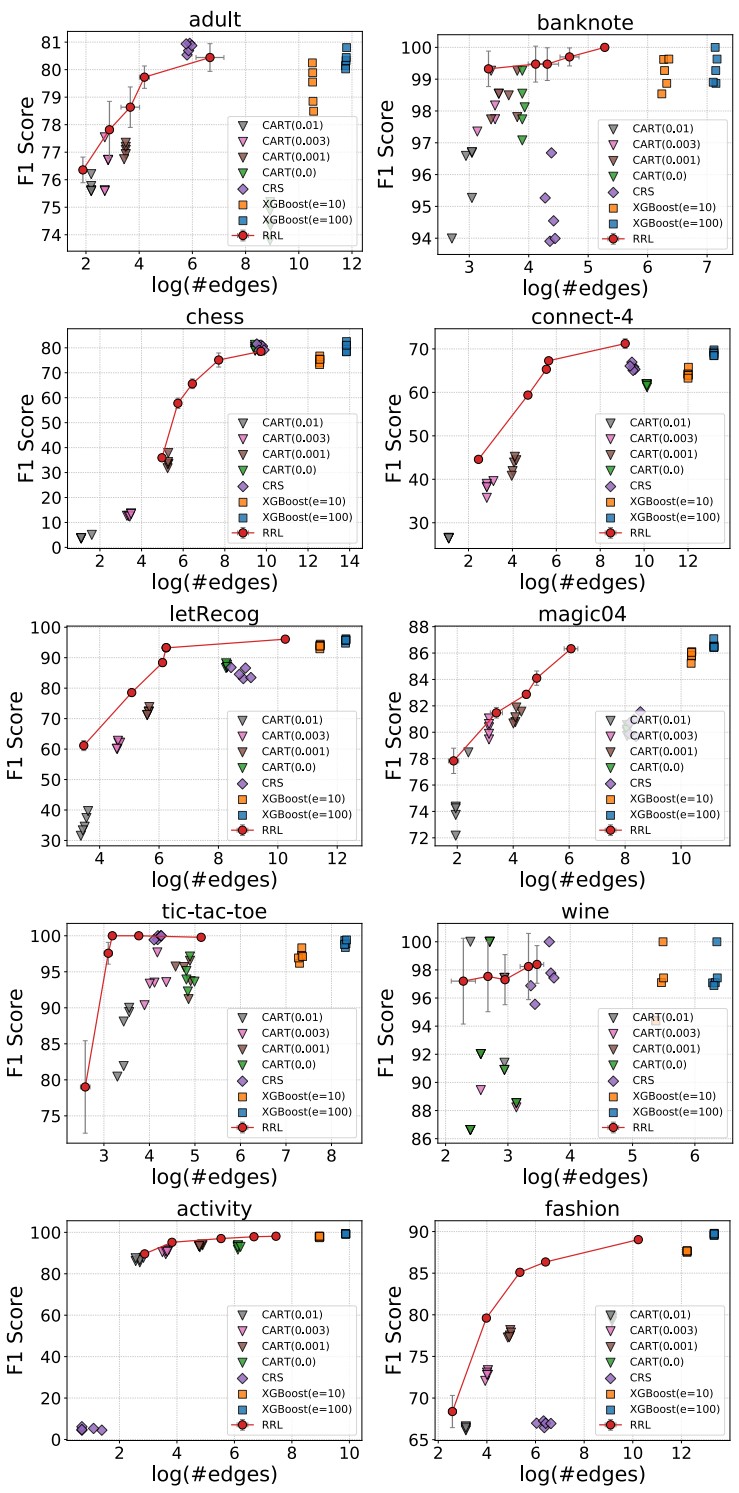

Figure 7: Scatter plot of F1 score against log(#edges) for RRL and baselines on ten datasets.