# OpenReview forum: "Scalable Rule-Based Representation Learning for Interpretable Classification"
_NeurIPS.cc/2021/Conference — NeurIPS 2021 Poster_

### Official Review · Reviewer_Uttc · 2021-07-01

**Rating:** 6
**Confidence:** 4

**Summary:**

The authors propose a new classifier - called Rule-based Representation Learner - that learns interpretable non-fuzzy rules. The model consists of 3 main modules:
- the binarization layer
- the logical layer(s)
- the linear layer.
Further, the authors propose a new training algorithm called Gradient Grafting.
The model was tested tested on 13 datasets and compared with six interpretable models and five complex models.

**Ethics Review Area:**

["I don’t know"]

**Limitations And Societal Impact:**

Yes

**Main Review:**

**Novelty**: novel. The authors propose a new method that automatically learns non-fuzzy rules for data representation and classification.

**Clarity**: clear. The paper is easy to follow.

**Quality**: the paper seems sound, however I have the following concerns:

- the authors propose Gradient Grafting, and their experiments show that it converges. However, the authors do not provide any intuition on why this new training method should work better than the others.

- The authors, while implicitly using them, never mention the t-norms. According to [1], a fuzzy logic can be defined upon a certain t-norm (triangular norm) representing an extension of the Boolean conjunction. A t-norm is a function $T : [0,1]^2 \to [0,1]$ that, for every $a,b,c \in [0,1]$, satisfies the following properties:
$$
T(a,b) = T(b,a),  \qquad
T(a,T(b,c)) = T(T(a,b),c),
$$
$$
a \le b \to T (a, c) \le T (b, c)  \qquad
T(a,1) = a, \qquad
T(a,0) = 0.
$$
So, if we consider the product t-norm we can represent:
$$
x \wedge y, \qquad x \cdot y,
$$
$$
\neg x, \qquad 1-x.
$$
$$
x \vee y = \neg (\neg x \wedge \neg y), \qquad x+y -xy = 1 - (1-x)(1-y)
$$
Which are exactly the functions used in Eq. (2) in the paper. The authors should add a discussion on the relationship between Eq. (2) and t-norms in order to justify why they model conjunction and disjunction in the proposed way. Further, they should discuss whether the improved logical activation functions retain the properties of the t-norms.

- Why do the authors need to add the small constant $\epsilon$ in Eq. 5?

- the authors mention the following "trick": '' One trick to deal with it is replacing $\frac{\partial \mathbb{P}(v)}{\partial v}$ with $\frac{P(P^2(v))}{v}$ for $P$ can lower the speed of approaching 0 while keeping the value range and tendency.'' Has this been tested?

- the authors compare the models by average rank, however, as proposed by Demsar himself in [2], when comparing multiple models models over multiple datasets the Friedman test + post-hoc test should be used to verify the statistical significance of the results.

- Parameter settings not specified. It would be highly beneficial to have a table stating exactly the number of layers and the number of neurons per layer for each dataset (and hence the final number of rules obtained). The interpretability of the model itself highly depends on this. Further, it would also help the reader to include in Table 4 in the appendix not only the average length of the rules, but also the minimum and maximum length of the rules.

- I am not sure I understand the graphs in Fig. 2. Regarding RRL: the authors plot how the F1-score varies according to the number of edges. Regarding the other models: what does the $x$ axis represent for the other models? For example, consider the bank-marketing dataset why is CART's F1-score plotted on left of the graph, while XGBoost F1-score gets plotted on the right?

[1] Petr Hajek. Metamathematics of fuzzy logic, volume 4. Springer Science & Business Media, 2013.

[2] Demsar, J. (2006). Statistical comparisons of classifiers over multiple data sets. Journal of Machine Learning Research, 7, 1–30.


**Time Spent Reviewing:**

4

---

> ### Author Response · Authors · 2021-08-08
> **Response to Reviewer Uttc**
>
> We appreciate the reviewer's effort. Here are our responses to the comments.
>
> >Comment 1: the authors propose Gradient Grafting, and their experiments show that it converges. However, the authors do not provide any intuition on why this new training method should work better than the others.
>
> As we mentioned in Section 3.3, the gradients of RRL at discrete points have no useful information in most cases (See Appendix E for the analysis). Therefore STE is not suitable for RRL. Other methods like ProxQuant and RB cannot directly optimize for the discrete model and be scalable at the same time. Gradient Grafting can directly optimize the loss of discrete models and use the gradient information at both continuous and discrete points, which overcomes the problems occurring in RRL training when using other gradient-based discrete model training methods.
>
> ---
>
> >Comment 2: The authors, while implicitly using them, never mention the t-norms. According to [1], a fuzzy logic can be defined upon a certain t-norm (triangular norm) representing an extension of the Boolean conjunction. The authors should add a discussion on the relationship between Eq. (2) and t-norms in order to justify why they model conjunction and disjunction in the proposed way. Further, they should discuss whether the improved logical activation functions retain the properties of the t-norms.
>
> We should clarify that one of our contributions is proposing the improved logical activation functions based on the logical activation functions proposed by Payani and Fekri (2019). We cite their paper in Section 3.1 (line 164). The relationship between Eq. (2) and t-norms is introduced in that paper (in Section 2.1). For the improved logical activation functions can keep the same range and tendency with logical activation functions (See line 187), they retain the properties of the t-norms. We will add the discussion about t-norms in our final version. Thanks for the suggestion.
>
> ---
>
> >Comment 3: Why do the authors need to add the small constant $\epsilon$ in Eq. 5?
>
> Because $\mathbb{P}(v)=\frac{-1}{-1+\log(v)}$, and $\epsilon$ is used to avoid $\log(0)$.
>
> ---
>
> >Comment 4: the authors mention the following "trick": '' One trick to deal with it is replacing $\frac{\mathbb{P}^2(v)}{v}$ with $\frac{\mathbb{P}(\mathbb{P}^2(v))}{v}$ for $\mathbb{P}$ can lower the speed of approaching 0 while keeping the value range and tendency.'' Has this been tested?
>
> Yes. As we mentioned in Appendix C (line 520), we use this trick for the RRL trained on the large data sets.
>
> ---
>
> >Comment 5: the authors compare the models by average rank, however, as proposed by Demsar himself in [2], when comparing multiple models models over multiple datasets the Friedman test + post-hoc test should be used to verify the statistical significance of the results.
>
> Thanks for the suggestion. We will add the results of the Friedman test and post-hoc test in our final version.
>
> For Table 1, the Friedman statistic is 73.837, corresponding to a p-value of $2.264 \times 10^{-11}$. Hence the null hypothesis of no significant differences is rejected. We then calculate pairwise comparisons using Conover post-hoc test. The results are shown in the following table.
>
> |	RRL vs|	C4.5|	CART|	SBRL|	CORESL|	CRS|	LR|	SVM|	PLNN(MLP)|	RF|	LightGBM|	XGBoost|
> |---|---|---|---|---|---|---|---|---|---|---|---|
> |	p-value|	0.00032|	0.00005|	0.00002|	0.00000|	0.00398|	0.00067|	0.00686|	0.04137|	0.07913|	0.97897|	0.95796|
>
> We can see that RRL significantly outperforms all the compared interpretable models with $p<0.001$.
>
> ---
>
> >Comment 6: Parameter settings not specified. It would be highly beneficial to have a table stating exactly the number of layers and the number of neurons per layer for each dataset (and hence the final number of rules obtained). The interpretability of the model itself highly depends on this.
>
> The parameter settings are listed in Appendix C. It should be noted that the final number of rules is not determined by the number of layers and the number of neurons per layer. It is determined by the edge connections of the discrete RRL after training. If one node (neuron) in the discrete RRL is not connected with any edge, it represents no rules. Considering that reused structures exist in rule-based models, we use the total number of edges instead of the total length or the total number of all rules as the metric of model complexity for rule-based models. Low model complexity indicates good model interpretability. The model complexities of RRLs trained on all 13 data sets are shown in Figure 2 and Figure 6. Figure 6 is in Appendix I.
>
> ---
>
> >Comment 7: it would also help the reader to include in Table 4 in the appendix not only the average length of the rules, but also the minimum and maximum length of the rules.
>
> We will include the minimum and maximum length of the rules in Table 4 in the appendix in our final version. Thanks for the advice.
>
> ---
>
> >Comment 8: I am not sure I understand the graphs in Fig. 2. Regarding RRL: the authors plot how the F1-score varies according to the number of edges. Regarding the other models: what does the $x$ axis represent for the other models? For example, consider the bank-marketing dataset why is CART's F1-score plotted on left of the graph, while XGBoost F1-score gets plotted on the right?
>
> Since CART is a decision tree model and XGBoost is an ensemble model consisting of many decision trees, we can use the total number of edges to evaluate their model complexities. Hence, the $x$ axis also represents the model complexity for the other models. Low model complexity indicates good model interpretability. The reason that CART's F1-score is plotted on the left of the graph while XGBoost's F1-score gets plotted on the right is that XGBoost needs tens or hundreds of decision trees to get that F1 score, so it is much more complex than CART. The right side of the graph has greater values of log(\#edges) than the left side.

---

### Official Review · Reviewer_WXKm · 2021-07-14

**Rating:** 6
**Confidence:** 4

**Summary:**

This submission proposes a new rule-based learner for classification. The learner uses discrete logical learners to obtain the rules. In order to obtain differentiability, the authors also propose a continuous approximation to the discrete layers. After a discussion about avoiding vanishing gradients problem, the authors conduct numerical experiments on a set of datasets.

**Limitations And Societal Impact:**

Yes they have.

**Main Review:**

The authors propose a new network learner using discrete and continuous layers. Although their main objective is to come up with an interpretable machine learning approach, they also obtain a method that performs remarkably well according to the numerical experiments. I believe the strength of the proposed method is not its interpretability but its accuracy as it beats (or performs quite close to) several sophisticated algorithms including SVM, XGBoost and Random Forest.

When it comes to the interpretability-accuracy trade-off, I have several questions and comments:

1) The number of obtained rules with the proposed approach is still quite large. Do you obtain less number of rules than the ones obtained with the interpretable models like CRS and CORELS?

2) Do the authors use the resulting set of rules (with the weights for classification) after training, or do they use the trained model with the continuous weights for classification? I guess this is related to whether they use the discrete or the continuous weights in the logical layers during testing.

3) How does the random bound selection in the binarization layer affect the interpretation? Given different bounds for two trained models on the same dataset, could the interpretations be completely different?

4) In Figure 2, the performance of CART is not good for bank-marketing dataset. But its performance looks quite good in Table 1. What is the difference?

5) The authors report the weights with absolute values. But clearly some of the weights could be negative. How can one interpret negative weights?

6) In Section 4.5, the authors state that "However, in practice, we can ignore these weight rules and only focus on rules with large weights first." However, in this case the accuracy of the resulting set of rules does not necessarily match the accuracy obtained with all the rules.  Therefore, this statement is not exactly true.

7) The submission claims to  propose a scalable rule-based learner but the related work considered in rule-based methods do not involve some recent work after 2017. For instance
Wei et al., (2019), Generalized linear rule models.
Lin et al., (2020) Generalized and scalable optimal sparse decision trees

Overall, the numerical experiments show that the authors have come up with a quite accurate algorithm. However, as I listed above, the arguments supporting the interpretability of the algorithm are not convincing.

**Time Spent Reviewing:**

4 hours

---

> ### Author Response · Authors · 2021-08-08
> **Response to Reviewer WXKm**
>
> We appreciate the reviewer's effort. Before responding to the reviewer's comments, we want to clarify that the reason we consider RRL interpretable is:
>
> (1) The model structure and components of RRL ensure it learns representation using discrete rules, so we can understand RRL like other interpretable rule-based models, e.g., decision trees, which is much easier than understanding the ordinary neural networks.
>
> (2) Experimental results show that RRL can obtain a better trade-off between classification performance and model complexity than other rule-based models. In other words, compared with other rule-based models, RRL can achieve close performance with better interpretability or achieve close interpretability with better performance.
>
> Here are our responses to the comments.
>
> >Comment 1: The number of obtained rules with the proposed approach is still quite large. Do you obtain less number of rules than the ones obtained with the interpretable models like CRS and CORELS?
>
> As shown in Figure 2 and Figure 6 (in Appendix I), we can easily obtain RRL with different numbers of rules (numbers of edges) by adjusting the hyperparameters of RRL. We can see that, compared with the interpretable models, i.e., CRS and CART, RRL can get a lower model complexity (less number of rules) in most cases when their F1 Scores are close. In other words, RRL can make better use of rules than other interpretable rule-based models.
>
> ---
>
> >Comment 2: Do the authors use the resulting set of rules (with the weights for classification) after training, or do they use the trained model with the continuous weights for classification? I guess this is related to whether they use the discrete or the continuous weights in the logical layers during testing.
>
> As mentioned in Section 3.1 (line 130) and Section 3.4 (line 241), we only use the discrete RRL for testing and interpretation. The results of RRL shown in the experiments are all from discrete RRL, i.e., the weights of the logical layers take discrete values (0 or 1).
>
> ---
>
> >Comment 3: How does the random bound selection in the binarization layer affect the interpretation? Given different bounds for two trained models on the same dataset, could the interpretations be completely different?
>
> If the number of the randomly generated bounds in the binarization layer is sufficient, the randomness will have no effect on the interpretation. As we mentioned in Section 3.2 (line 212), the weights of logical layers are trainable, which means we can choose bounds in an end-to-end way. Therefore, although the randomly generated candidate bounds could be different each time, the logical layer will select the similar bounds by edge connections to minimize the training loss during the training. Hence, if the training loss converges, the two RRL models trained on the same data set will be very similar even if the given candidate bounds are different.
>
> ---
>
> >Comment 4: In Figure 2, the performance of CART is not good for bank-marketing dataset. But its performance looks quite good in Table 1. What is the difference?
>
> In Figure 2, we report the classification performance of CART with different model complexities. Different colors for the same legend symbol (e.g., triangle for CART) indicate different model complexities (see line 308). On the bank-marketing data set, the F1 Scores of the grey and pink triangles are not good because their complexities are too low to fit the data well. The brown and green triangles can get similar performance as the results shown in Table 1 since their complexities are sufficient for the task.
>
> ---
>
> >Comment 5: The authors report the weights with absolute values. But clearly some of the weights could be negative. How can one interpret negative weights?
>
> A negative weight of a rule w.r.t. a certain class indicates that if the corresponding rule is satisfied by an instance, the probability of the instance belonging to the corresponding class will be low. For example, the rules shown in Figure 4b have positive weights of the class "Yes" and negative weights of the class "No" (we did not show them in Figure 4b). The negative weights of the class "No" mean that clients satisfying these rules are less likely to reject the telesales.
>
> ---
>
> >Comment 6: In Section 4.5, the authors state that "However, in practice, we can ignore these weight rules and only focus on rules with large weights first." However, in this case the accuracy of the resulting set of rules does not necessarily match the accuracy obtained with all the rules. Therefore, this statement is not exactly true.
>
> In our paper, after the sentence the reviewer quoted (line 349), we write "After analyzing rules with large weights and having a better understanding of the learned RRL and the data set, we can then understand those less important rules gradually." Therefore, we do not mean we should ignore the rules with small weights all the time. We try to say that understand the rules with large weights first is a better way to understand the whole model when the model is complex.
>
> ---
>
> >Comment 7: The submission claims to propose a scalable rule-based learner but the related work considered in rule-based methods do not involve some recent work after 2017. For instance Wei et al., (2019), Generalized linear rule models. Lin et al., (2020) Generalized and scalable optimal sparse decision trees
>
> Thanks for the suggestions. We will discuss these two works in Section 2 in our final version.
>
> Wei et al., (2019) use column generation (CG) to search and produce candidate rules, and a generalized linear model (GLM) is re-fit as rules are generated. However, CG is solved using either integer programming (IP) or a heuristic algorithm. The scalability of IP is not good, while the heuristic algorithm could affect model accuracy.
>
> Hu et al., (2019) propose OSDT that combines strong analytical bounds that reduce the search space with custom data structures and computational caching to produce algorithms to find optimal decision trees. Lin et al., (2020) propose GOSDT based on OSDT to support a collection of additional objective functions and a novel dynamic programming with bounds algorithm. Although their bounds are useful, these models could take too much time when dealing with data with hundreds or thousands of features since the search space is still very large.

---

### Official Review · Reviewer_v1eM · 2021-07-18

**Rating:** 7
**Confidence:** 4

**Summary:**

The paper introduces a gradient-descent and network-based training method to learn a rule-based interpretable classifier by training concurrently a continuous model and a discretized version.

**Ethical Concerns:**

None.

**Limitations And Societal Impact:**

The authors have not addressed this question, nor would I see how they could have done that.

**Main Review:**

A discrete model and a continuous one are trained concurrently, but the way the gradient information by the continuous model used to affect (support to train) the discrete model is not presented clearly.

Using k random upper bounds and k random lower bounds for each continuous attribute in binarization step maybe is not a good practice, since the bounds are selected randomly and continuous attributes can be different in ranges and number of distinct values. The hyper-parameter k is added to the set of inherent hyper-parameters of the network-based training methods that makes the algorithm more difficult to apply.

The RRL algorithm is stated to be scalable; however how its efficiency (running time) and consumed memory are that is not reported. The following features can make the algorithm less scalable in term of running time.
- To support processing directly continuous attributes, the way of binarization used by the RRL increases very much the dimension of input vectors (at Binarization layer).
- To support both CNF and DNF, RRL algorithm maintains concurrently Conjuction part and Disjunction part in each Logical layer that also makes Logical layer bigger.
- A discrete model and a continuous one are trained concurrently.

How is the behavior of training loss of Gradient Grafting on the other 11 datasets, it could be nice if its training loss for the other 11 dataset is supplemented in Figure 3.

A model with high classification performance in F1-score does not imply it will be high in Accuracy metric. Choosing Accuracy or F1-score metric to evaluate classification performance of a model depends on that (TP, TN) or (FP, FN) is more important in a particular application rather than on whether the class distribution is balance or not. It could be more interesting to report the performance in accuracy metric to see how the ranking will be.


**Time Spent Reviewing:**

4 hours

---

> ### Author Response · Authors · 2021-08-08
> **Response to Reviewer v1eM**
>
> We appreciate the reviewer's effort. Here are our responses to the comments.
>
> >Comment 1: A discrete model and a continuous one are trained concurrently, but the way the gradient information by the continuous model used to affect (support to train) the discrete model is not presented clearly.
>
> As we mentioned in Section 3.1 (line 158) and Figure 1b, the discrete model and the continuous model actually share the same parameters, but the discrete model needs to discretize the parameters first. Therefore, the update on the parameters of the continuous model also affects the parameters of the discrete model.
>
> ---
>
> >Comment 2: a) Using k random upper bounds and k random lower bounds for each continuous attribute in binarization step maybe is not a good practice, since the bounds are selected randomly and continuous attributes can be different in ranges and number of distinct values. b) The hyper-parameter k is added to the set of inherent hyper-parameters of the network-based training methods that makes the algorithm more difficult to apply.
>
> a) If the number of the randomly selected bounds, i.e., 2k, is insufficient, the performance may be affected when the ground truth bounds for the continuous feature are unevenly distributed. However, our experiments on eight data sets with continuous features show that binarization layers with k no more than 50 can deal with the continuous features well. Therefore, the randomly generated bounds are effective in most cases. When the values of one continuous feature are very unevenly distributed, we can use k bins to generate the candidate bounds. For example, values in each bin have the same nearest center of a 1D k-means cluster. Moreover, we are now trying to make the bounds trainable. However, the trainable bounds make the whole model unstable and hard to train.
>
> b) It is hard and could make the binarization layer very complex to implement if we try to remove the hyperparameter k (i.e., a trainable k). Additionally, the hyperparameter k is easy to tune since we can decrease it safely if the performance does not change. Moreover, RRL is not sensitive to k as long as k is not too small.
>
> We will try to find a better alternative to the current binarization layer in future work.
>
> ---
>
> >Comment 3: The RRL algorithm is stated to be scalable; however how its efficiency (running time) and consumed memory are that is not reported.
>
> The training time of RRL is shown in Table 3 in Appendix F. As we mentioned in Appendix C, all the experiments are conducted with one GeForce RTX 2080 Ti GPU. Therefore, the consumed GPU memory is no more than 11GB.
>
> ---
>
> >Comment 4: The following features can make the algorithm less scalable in term of running time.
> >
> >a) To support processing directly continuous attributes, the way of binarization used by the RRL increases very much the dimension of input vectors (at Binarization layer).
> >
> >b) To support both CNF and DNF, RRL algorithm maintains concurrently Conjuction part and Disjunction part in each Logical layer that also makes Logical layer bigger.
> >
> >c) A discrete model and a continuous one are trained concurrently.
>
> a) If we set the number of bounds in the binarization layer, i.e., 2k, too large, it indeed could affect the scalability of our model. Our experimental study has shown that it is usually sufficient to set k to 50 (10 or 20 is enough for most cases).
>
> b) If we have no idea about the data and task, then the conjunction and disjunction layers are all necessary. If we already know that, for example, DNF is more suitable for the task, we can decrease the number of nodes corresponding to CNF.
>
> c) For each step, RRL needs two forward and one backward propagation, while ordinary neural networks need one forward and one backward propagation. Compared with ordinary neural networks, each step of RRL only needs an additional forward propagation (for the discrete model). The forward propagation for the discrete model is actually fast (we can disable the gradient calculation because of the Gradient Grafting). Hence, its effect on scalability is very small.
>
> ---
>
> >Comment 5: How is the behavior of training loss of Gradient Grafting on the other 11 datasets, it could be nice if its training loss for the other 11 dataset is supplemented in Figure 3.
>
> The behaviors of training loss of Gradient Grafting shown in Figure 3 are representative examples, and the behaviors on the other 11 data sets are similar. We can add the training loss for other data sets in the appendix in our final version.
>
> ---
>
> >Comment 6: A model with high classification performance in F1-score does not imply it will be high in Accuracy metric. Choosing Accuracy or F1-score metric to evaluate classification performance of a model depends on that (TP, TN) or (FP, FN) is more important in a particular application rather than on whether the class distribution is balance or not. It could be more interesting to report the performance in accuracy metric to see how the ranking will be.
>
> Using F1 Score (Macro) as the metric can avoid some evaluation issues on imbalanced data sets. Specifically, if one model just predicts the most popular class, it will get a very low F1 Score (Macro). However, if one model just predicts the most popular class, it can still get a not bad accuracy, especially on the imbalanced data sets. In our experiments, there are some imbalanced data sets. For example, the percent of the most popular class is 76\% and 88\% for the adult data set and the bank-marketing data set, respectively. Therefore, we use the F1 Score (Macro) instead of Accuracy as the metric of classification performance. We will report the performance using accuracy in the appendix of our final version.

---

### Official Review · Reviewer_nbhN · 2021-07-19

**Rating:** 6
**Confidence:** 4

**Summary:**

This paper presents a new scalable classifier, named Rule-based Representation Learner (RRL), that can automatically learn interpretable rules for data representation and classification.  A new gradient-based discrete model training method, i.e., Gradient Grafting, is proposed as well that directly optimizes the discrete model. RRL's performance on structured data seems promising wrt experiments performed.




**Ethical Concerns:**

This paper can be found in https://openreview.net/forum?id=UwOMufsTqCy as an ICLR 2021 submission that was rejected. If authors are same, then no issue, else plagiarism as per Docoloc is 42%. See:- https://www.docoloc.de/d88c4d5162dd4b21eeb6add446e2b5bbCEuFHG85rulAsphkx9/en/konto.hhtml?dogetresult=49. Please crosscheck this if its fine for a green signal. I assume it's fine.

**Ethics Review Area:**

["Research Integrity Issues (e.g., plagiarism)"]

**Limitations And Societal Impact:**

There is no issue wrt negative Societal Impact, hence skipped.

**Main Review:**

Novelty:
1. Although influenced by Wang et al [2020], the proposed gradient grafting technique based RRL is novel as it can handle vanishing gradient and large no of features.

Strengths:
1. Code available at https://github.com/anonymous-rrl/rrl
2. The layers are well explained.
3. The proposed gradient grafting technique seems to work quite well in the specified experiments.
4. Case study proves application in practical scenario.

Weakness:
1. Explanation on 'interpretability' part is missing while accuracy is established. Using rules increases expectation interpretation.
2. Experiments are sound, however the worst accuracy obtained should have been reported as well across datasets along with corner cases of failures.
3. Limitations of the paper (unstructured data) are only mentioned as one liner in Section 5. This needs expansion as a subsection including implementation level and formulation level limitations and assumptions.

Relevance to NeurIPS:
This paper is relevant to Neural Information Processing System conference, as we are moving slowly towards Neuro-symbolic AI papers.

Correctness:
Claimed formulations and results seem to be correct.

Clarity:
Paper is well written.

Related work Comparison:
Comparison is fine wrt state-of-the-art.

Effort: The amount of work done in writing paper, coding, experimenting, supplementary material is appreciable.

Suggestions:
1. It is better to do a Train-Eval-Test split to avoid biased results following guidelines in Andrew Ng's "Machine Learning Yearning" book.
2. Try to bring in the mathematical guarantees, upper-lower bound of the testing stage.
3. Apart from Figure 1 inner blocks, a pipeline of the whole process would have made the picture clear in a single figure.

Verdict: This paper falls in the above acceptance threshold area and there are still some concerns in terms of analysis of results in terms of interpretability. Differentiation with related work was expected to be more rigorous. However, this is a good work. Papers on rule based learning are rare wrt the general bulk of usual deep learning papers - so it is recommended to accept the paper to maintain flavour of NeurIPS, provided other reviewers agree.

**Needs Ethics Review:**

Yes

**Time Spent Reviewing:**

3

---

> ### Author Response · Authors · 2021-08-08
> **Response to Reviewer nbhN**
>
> We appreciate the reviewer's effort. Here are our responses to the comments.
>
> >Weakness 1: Explanation on 'interpretability' part is missing while accuracy is established. Using rules increases expectation interpretation.
>
> As we mentioned in Section 4.3, interpretability and classification accuracy are both important for interpretable models. Therefore, what we really care about is the relationship between interpretability and accuracy. Since RRL is a rule-based model, the model complexity is the main factor affecting model interpretability. Hence, we show the relationship between model complexity and F1-score of both RRL and baselines in Figure 2 and Figure 6 (in Appendix I). We can see that RRL can obtain a better trade-off between model complexity and classification performance than other rule-based models. In other words, compared with other rule-based models, RRL can achieve close classification performance with better interpretability or achieve close interpretability with better classification performance.  Additionally, we introduce how to interpret RRL in Section 3.4 and use the case study in Section 4.5 to further verify the good interpretability of RRL.
>
> ---
>
> >Weakness 2: Experiments are sound, however the worst accuracy obtained should have been reported as well across datasets along with corner cases of failures.
>
> In Table 1, we can observe that RRL performs well on almost all the data sets, but its result on the chess data set is not so good. The reason is that we need a quite complex model to fit the chess data set well (as the results shown in Figure 6), and rules in DNF could be more suitable for this data set (as revealed by the performance of CRS). Since the numbers of nodes in conjunction layer and disjunction layer in each logical layer are set to the same in our experiment, the number of nodes corresponding to rules in DNF is equal to the number of rules in CNF. Therefore, only half of the nodes correspond to rules in DNF. When the data set needs a large number of rules in DNF and the number of nodes in the logical layer is restricted,  the performance of RRL could be affected. The solution is simple since we can change the ratio of nodes in conjunction layer to nodes in disjunction layer in each logical layer to increase the number of nodes corresponding to rules in DNF.
>
> ---
>
> >Weakness 3: Limitations of the paper (unstructured data) are only mentioned as one liner in Section 5. This needs expansion as a subsection including implementation level and formulation level limitations and assumptions.
>
> Thanks for the advice. The limitation of our paper is that the current design of RRL is limited to structured data. When dealing with unstructured data using RRL, we need to convert them into structured data first. For example, we need to convert the 2D image into a 1D vector when using RRL, which could lose spatial information. We can compare RRL to MLP, and it is promising to design new models with different structures or components based on RRL that can deal with specific data types, like CNN or LSTM. We will introduce the limitation of this paper in more detail in our final version.
>
> ---
>
> >Suggestion 1: It is better to do a Train-Eval-Test split to avoid biased results following guidelines in Andrew Ng's "Machine Learning Yearning" book.
>
> In this paper, we adopt 5-fold cross-validation to evaluate the classification performance and avoid biased results. When parameter tuning is required, 95\% of the training set was used for training and 5\% for validation.
>
> ---
>
> >Suggestion 2: Try to bring in the mathematical guarantees, upper-lower bound of the testing stage.
>
> Thanks for the suggestion. Although it is hard to derive the mathematical guarantees, they could be useful to help us further understand our model. This is also one direction of our future work.
>
> ---
>
> >Suggestion 3: Apart from Figure 1 inner blocks, a pipeline of the whole process would have made the picture clear in a single figure.
>
> Combining Figure 1b and Figure 1a, we can obtain the pipeline of the whole process. Figure 1b tells us how the data flows between models, while Figure 1a tells us how the data flows between layers.

---

### Review · Ethics_Reviewer_enQy · 2021-08-09

**Recommendation:** process for potential code of conduct…

**Ethics Review:**

the plagiarism concern raised by a reviewer is a potential code of conduct violation separate from the NeurIPS ethical guidelines

---

### Review · Ethics_Reviewer_x9pu · 2021-08-13

**Recommendation:**

The ethical review was flagged to ensure the paper was not plagiarizing a previously submitted (but rejected) ICLR paper. As I cannot see the author names, i cannot verify.

Please have the appropriate party review.

**Ethics Review:**

The ethical review was flagged to ensure the paper was not plagiarizing a previously submitted (but rejected) ICLR paper. As I cannot see the author names, i cannot verify.

Please have the appropriate party review.

---

### Decision · Program_Chairs · 2021-09-27

**Decision:**

Accept (Poster)

**Comment:**

This paper presents a new scalable classifier, named Rule-based Representation Learner (RRL), that can automatically learn interpretable rules for data representation and classification. A new gradient-based discrete model training method, i.e., Gradient Grafting, is proposed as well that directly optimizes the discrete model. RRL's performance on structured data seems promising wrt experiments performed.

Overall, the reviewers agree that there is merit in the proposed approach and they all support the acceptance of the manuscript. I also looked at aspects of the manuscript and found the experimental evaluation very well organized and thorough. Moreover, I particularly like the results properly showing the trade-offs of complexity and performance and the comparison with other methods according to these two measures. This can convincingly, and in a quantitative manner show, that the proposed method can produce short explanations (i.e., small number of rules/edges) while still performing relatively well. Moreover, when allowing complex explanations, the model performs provides state-of-the-art results when compared to other approaches that produce less interpretable predictors. I believe that approaches like the proposed ones will be important to get AI accepted with application fields, such as medicine or finance.

In summary, there are strong arguments supporting the acceptance of the manuscript and I recommend accepting the manuscript.